# Using spatio-temporal graph neural networks to estimate fleet-wide photovoltaic performance degradation patterns

**Yangxin Fan[2]\*, Raymond Wieser[1], Xuanji Yu[1], Yinghui Wu[2], Laura S. Bruckman[1], Roger H. French[1]**

**1** Department of Materials Science and Engineering, Case Western Reserve University, Cleveland, Ohio, United States of America, **2** Department of Computer and Data Sciences, Case Western Reserve University, Cleveland, Ohio, United States of America

\* yxf451@case.edu

**Data Availability Statement:** We have uploaded our work as a publicly available package on the PyPi python package repository. This includes the model framework and the scripts used to generate

## Abstract

Accurate estimation of photovoltaic (PV) system performance is crucial for determining its feasibility as a power generation technology and financial asset. PV-based energy solutions offer a viable alternative to traditional energy resources due to their superior Levelized Cost of Energy (LCOE). A significant challenge in assessing the LCOE of PV systems lies in understanding the Performance Loss Rate (*PLR*) for large fleets of PV systems. Estimating the *PLR* of PV systems becomes increasingly important in the rapidly growing PV industry. Precise *PLR* estimation benefits PV users by providing real-time monitoring of PV module performance, while explainable *PLR* estimation assists PV manufacturers in studying and enhancing the performance of their products. However, traditional *PLR* estimation methods based on statistical models have notable drawbacks. Firstly, they require user knowledge and decision-making. Secondly, they fail to leverage spatial coherence for fleet-level analysis. Additionally, these methods inherently assume the linearity of degradation, which is not representative of real world degradation. To overcome these challenges, we propose a novel graph deep learning-based decomposition method called the Spatio-Temporal Graph Neural Network for fleet-level *PLR* estimation (PV-stGNN-PLR). PV-stGNN-PLR decomposes the power timeseries data into aging and fluctuation components, utilizing the aging component to estimate *PLR*. PV-stGNN-PLR exploits spatial and temporal coherence to derive *PLR* estimation for all systems in a fleet and imposes flatness and smoothness regularization in loss function to ensure the successful disentanglement between aging and fluctuation. We have evaluated PV-stGNN-PLR on three simulated PV datasets consisting of 100 inverters from 5 sites. Experimental results show that PV-stGNN-PLR obtains a reduction of 33.9% and 35.1% on average in Mean Absolute Percent Error (MAPE) and Euclidean Distance (ED) in *PLR* degradation pattern estimation compared to the state-of-the-art *PLR* estimation methods.

the data used in training the model. Package: https://pypi.org/project/PVplr-stGNN/#description We have also used data avaiilible on OSF.io to further benchmark the performance of our models, and the data we used is avaiilible here: OSF Data: https://osf.io/8ucaq/.

**Funding:** Y.F, R.W, X.Y, Y.W, L.S.B,F.H.F This material is based upon work supported by the U.S. Department of Energy's Office of Energy Efficiency and Renewable Energy (EERE) under Solar Energy Technologies Office (SETO) Agreement Number DE-EE0009353. The funders had no role in study design, data collection and analysis, decision to publish, or preparation of the manuscript.

**Competing interests:** The authors have declared that no competing interests exist.

# 1 Introduction

Photovoltaic energy represents a promising solution to growing uncertainty over the stability of the world's energy resources and lowering the carbon footprint. As photovoltaics (PV) have become an increasingly dominant energy sector over past few decades [1], the estimation of Performance Loss Rate (*PLR*) of PV systems has become a recent focus of research [2–4]. Such estimations have the potential to provide PV manufacturers with a competitive edge by allowing them to estimate system performance without relying on time-consuming experiments or consistent monitoring. As the PV market continues to grow, there is a demand for degradation analysis of a large number of PV systems, which exhibit spatio-temporal correlations that can enhance model performance.

While PV has become a well-established technology, large-scale deployment and adoption of PV power generation depends on emphasizing the financial benefits of investing in commodity scale solar. This is best accomplished through careful accounting of all possible costs associated with the production of energy. The Levelized Cost of Energy (LCOE) is a measurement of the total cost of the generation of one unit of energy, and is used to quantify the profitability of a technology. The LCOE summarizes the lifetime of the system, including initial cost for planning, building, maintenance, and destruction of the site. A significant portion of the LCOE of a PV system is related to its ability to produce the same amount of power under the same weather conditions (*Performance Loss*) [5]. Performance loss in PV modules on an individual's roof may impact a single family electricity bill, but the performance loss across fleets of PV power plants contributes to millions of dollars in lost revenue. The *Performance Loss Rate* (*PLR*) of a PV system indicates the decline of the power output over time (in units of % per annum (%/a, or %/year)) [6]. *PLR* represents the physical degradation of PV modules, which is a unavoidable property of any PV system.

There are different ways to numerically characterize *PLR*. The *PLR* degradation pattern can be quantified by the performance of PV systems under the standard test conditions (STC) over time [7] or by the power produced for a given operating environment. The relative *PLR* [8] is calculated from power data compared to the initial state of the system, having units of %/annum (or %/a). The other is absolute *PLR* [9], which is the coefficient of the slope of a linear estimation of the degradation pattern. However, these two definitions assume a linear degradation rate, which cannot characterize diversified degradation patterns in real-world PV systems [5]. Additionally, traditional *PLR* methods often generate inconsistent and biased results due to the interplay of user decisions on a variety of modeling steps [6], and most often require accurate meteorology data for each system being model resulting in a large data overhead for PV Fleet monitoring. Furthermore, traditional *PLR* estimation methods typically assume systems exhibit linear degradation with a monotonically decreasing trend, which may not accurately represent the real-world degradation patterns of PV systems and does not account for transient conditions such as soiling, LID, LeTID, etc. . . This leads to erroneous descriptions of the performance of PV systems over their lifetimes for methods using aforementioned assumptions [5], which can mislead performance monitoring teams to over or underestimate the value proposition of their systems. An accurate understanding of the *PLR* of PV systems is critical to inform the LCOE calculation towards more accurate financial predictions, and allow for targeted assets maintenance [10].

Several computational challenges have been observed in *PLR* estimation. Conventional methods [11–14] rely on domain-specific knowledge, requiring manual tuning, site specific meteorology data and ad-hoc decisions, which hinder large-scale *PLR* analysis. Diversified degradation patterns, including non-linear and non-monotonic behavior, cannot be accurately captured by methods assuming monotonic aging [15, 16]. Traditional methods such as 6K,

PVUSA, XbX, and XbX+UTC [11–14] are limited to standalone analysis of individual PV systems, require complete meteorology data and cannot leverage spatial coherence for large-scale fleet-level analysis [6].

Inspired by the success of emerging neural networks and spatiotemporal graph models in PV research [17–20], we propose our novel approach called PV-stGNN-PLR, a meteorology agnostic *PLR* estimation model. Previous methods have modeled the PV timeseries datasets as a series of spatio-temporal (st)-graphs. Compared to previous stand-alone methods, st-graph-based methods exploit spatio-temporal and network topological features to better learn PV fleet-level representations, and outperform state-of-the-art imputation and forecasting methods. Our proposed st-graph-based PV-stGNN-PLR model effectively captures the long-term aging behaviors of PV systems in large fleets by treating the *PLR* estimation problem as an unsupervised machine learning task. PV-stGNN-PLR introduces high expressiveness with multiple Graph Autoencoder (GAE) based modules. A clear disentanglement among GAE outputs is achieved to distinguish aging and fluctuation representations. PV-stGNN-PLR also leverages the stationary nature of power timeseries in ideal PV systems and decomposes the measurements into a long-term degradation trend and fluctuation terms that account for seasonalities and noise.

The PV-stGNN-PLR model addresses the challenges of *PLR* estimation by utilizing Neural Networks and spatiotemporal graph modeling. PV-stGNN-PLR produces more consistent results with minimal user knowledge and without meteorology data. Our experiments have shown that PV-stGNN-PLR achieves better results compared to state-of-the-art traditional *PLR* estimation methods while requiring significantly less input. Overall, our approach improves the accuracy of *PLR* estimation, facilitating more precise LCOE calculations and enabling targeted maintenance strategies for PV systems.

## 2 Methods

In ideal PV systems without degradation, after removing the variations due to weather conditions across different seasons and years, long-term power timeseries should be stationary, Power timeseries measurements can be deconstructed into two major components: a long-term "degradation" trend that represents the changes ("up" and "down") in PV performance, and one or multiple fluctuation terms to capture one-level or multiple-level seasonalities and noises that correspond to each particular level of seasonal resolution. The high frequency signals exhibit spatial coherence, i.e locations that are in the same regions experience similar fluctuations at the roughly same time. PV-stGNN-PLR takes advantage of these facts along with leveraging large fleets of systems with diverse spatio resolution to better parse out the high frequency signals from the long term trend.

### 2.1 Proposed PV-stGNN-PLR framework

Existing spatio-temporal graph neural networks (stGNNs) are mostly designed for short-term predictive regression analysis [17–20]. Little has been done for leveraging spatio-temporal coherences learned by stGNNs to conduct long-term trend analysis.

Given the raw PV power timeseries data from a PV fleet, it is modeled as a series of st-graphs $G = \{G_1, \ldots, G_t, \ldots, G_T\}$, where all the st-graphs have the same set of nodes and share the same adjacency matrix $A$, with varying PV measurements at each node over time. The degradation pattern then is interpreted as the low frequency trend, known as "aging", of the timeseries data. The estimated degradation pattern is determined after removing out the fluctuation terms that capture high frequency noise and yearly seasonalities. Our proposed PV-stGNN-PLR, a graph neural network-based model, fits the "hidden" *PLR* (long-term

trend) pattern by minimizing a loss function that quantifies two major types of errors: reconstruction error of the aging term and the fluctuation terms. Additional terms have been added for smoothness and flatness regularization. The methodology is analogous to tradition time-series decomposition but embedded into graph structures using neural networks to decompose the individual components. We also implemented regression analysis on the extracted aging to derive the global *PLR* value as defined in the Eq 7.

**2.1.1 Modeling a PV fleet as a st-graph.** We model a PV fleet as a spatiotemporal graph (st-Graph) $G = (V, E, X_t)$, (1) nodes $V$ represent PV systems (inverters); (2) $E$ are the edges created by thresholded gaussian kernels from Eq 1 to connect spatially correlated PV systems together; and (3) $X_t \in \mathbb{R}^{N \times d}$ indicates node attributes with each node carrying a tuple of length $d$ recording $d$ measurements at timestamp t. We denote the total number of timestamps as $T$, the number of nodes as $N$. Since the locations of PV systems are fixed, the graph structure is static with time-invariant nodes and edges. However, $X_v(t)$ is time-varying. Each node carries a single power output measurement in our study ($d$ is 1).

$$A_{i,j} = \begin{cases} 1, i \neq j \text{ and } \exp(-\frac{dist_{ij}^2}{\sigma^2}) \geq \epsilon \\ 0 \text{ otherwise} \end{cases} \tag{1}$$

Here $A$ is the adjacency matrix of $G$. $dist_{ij}$ is the Euclidean Distance between the node pair $(i, j)$. $\sigma$ is the standard deviation of the distances. $\epsilon$ determines the network sparsity: the larger, the sparser the network is. Note that $\epsilon = 0$ indicates a "clique", *i.e.*, all nodes are pairwisely connected.

**2.1.2 PV-stGNN-PLR architecture.** The input of PV-stGNN-PLR is PV power timeseries data from a large PV fleet, modeled as a series of st-graphs. Each node represents a PV system that carries one power output timeseries. The model PV-stGNN-PLR consists of $k + 1$ paralleled GAEs that decompose input PV signals into one *aging term* and $k$ *fluctuation terms*. (1) The first module (denoted as $GAE_1$) extracts, encodes, and learns an aging representation $h_a$. (2) Each module $GAE_i$ ($i \in [2, k + 1]$) extracts a different fluctuation term. Each GAE module consists of an encoder and decoder.

*Encoder and decoder*. As illustrated in Fig 1, each encoder or decoder consists of a graph transform operator followed by a graph attention operator to form two convolutional or deconvolutional layers. This design derives its convention from established spatio-temporal

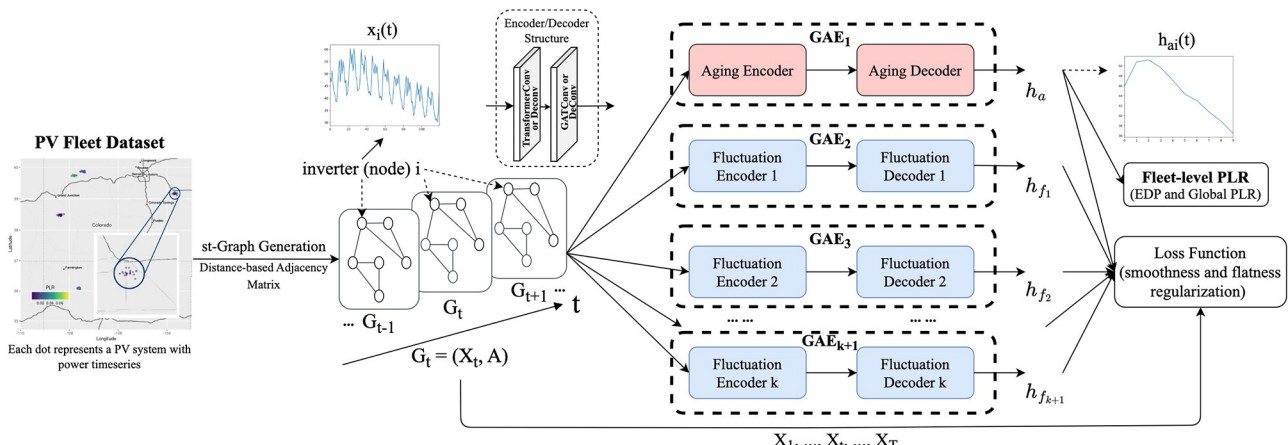

**Fig 1. Overview of PV-stGNN-PLR framework for fleet-level *PLR* estimation.** Map data provided by ©OpenStreetMap Contributors.

GNNs *e.g.*, [21]. We adapted this design for the basic module in PV-stGNN-PLR for aging and fluctuate representations for long-term trend analysis.

*Graph transformer operators.* These operators perform attentive information propagation between nodes by leveraging multi-headed attention into graph learning [22]. Multi-headed attention matrices replace the original normalized adjacency matrix as a transition matrix for message passing. Therefore, each node can effectively aggregate features information from its neighbors.

*Spatio-temporal graph attention operators.* We adapted the attention mechanism from graph attention networks (GATs) [23]. The operator uses masked self-attention layers and enable different weights to different nodes in a neighborhood by stacking layers, where nodes are able to attend over their neighborhoods' features. This makes PV-stGNN-PLR more sensitive to useful information from spatial and temporal "neighbors" in the PV networks.

PV-stGNN-PLR introduces high expressiveness with multiple GAE-based modules. A clear disentanglement among GAE outputs is achieved to distinguish aging and fluctuation representations. The extracted aging terms can estimate the degradation patterns of PV systems in a large PV fleet.

**2.1.3 PV-stGNN-PLR learning objective.** In order to properly parse out the different signals in the PV timeseries, a novel loss function was designed. The loss function of PV-stGNN-PLR consists of a global reconstruction error with additional constraints applied individually to the fluctuation and aging terms. Our novel design of the loss function achieves a separation between aging and fluctuation for long-term trend analysis for *PLR* estimation.

*Reconstruction error.* We aimed to decompose $X$ into aging term $h_a$ and $k$ different fluctuation terms $h_{f_1}, h_{f_1}, \ldots, h_{f_k}$ using $k + 1$ GAEs. To ensure the quality of decomposition, we wanted to minimize reconstruction error.

$$\min||X - h_a - \sum_{q=1}^{k} h_{f_q}||^2 \tag{2}$$

To ensure stationarity of the fluctuation part, we imposed two constraints on $h_f$: (1) mean constraint: we segmented every timeseries to ensure difference of mean between each segment within the series being as small as possible, and (2) slope constraint: the sum of absolute value of global slope of extracted $h_f$ for each node should be as close to zero as possible.

*Mean constraint.* For mean constraint, we partitioned $h_f$ of each node into $p$ segments each with length $w$. The length of $w$ determines the temporal resolution of extract fluctuation term. We then minimized the sum of difference between mean values of each pair of the segments to ensure flatness of the fluctuation over time as:

$$\min \sum_{q=1}^{k} \sum_{l=1}^{N} \sum_{i,j=1}^{p} (m_{q_{l_i}} - m_{q_{l_j}})^2 W_{ij} \tag{3}$$

Here (1) $m_{q_{l_i}}$, $m_{q_{l_j}}$ denotes the mean of the *i*-th / *j*-th segment of the *l*-th node's *q*-th fluctuation term, (2) $W$ is a weight matrix, where each entry $W_{ij}$ denotes a learnable weight to minimize the mean difference between segmented pair $m_{l_i}$ and $m_{l_j}$. To ensure the stationarity of the fluctuation terms, the long-term mean was minimized by applying linear growth of the weight based on the distance between $m_{l_i}$ and $m_{l_j}$. This is based on the following intuition: the farther the two segments are from each other, the more weights are given to minimize their mean difference.

*Slope constraint.* The slope constraint ensures the global flatness of each extracted fluctuation level, and is defined as:

$$\min \sum_{q=1}^{k} \sum_{l=1}^{N} |slope(h_{f_{q_l}})| \tag{4}$$

Here the slope is calculated from the least square fit.

*Smoothness constraint.* We also wanted to reduce the noise level in $h_a$ by minimizing sum of the standard deviation of the first-order differences of aging term of all the nodes. We thus introduced a smoothness term as:

$$\min \sum_{l=1}^{N} SD(h_{a_l}[t+1] - h_{a_l}[t]) \tag{5}$$

where $t \in [1, T-1]$ with $T$ the total number of $h_{a_l}$ timestamps, and SD denotes standard deviation.

*Loss function.* Putting these together, we formulated the loss function of PV-stGNN-PLR as:

$$\min ||X - h_a - \sum_{q=1}^{k} h_{f_q}||^2 + \lambda_1 \sum_{q=1}^{k} \sum_{l=1}^{N} \sum_{i,j=1}^{p} (m_{q_{l_i}} - m_{q_{l_j}})^2 W_{ij} +$$

$$\lambda_2 \sum_{q=1}^{k} \sum_{l=1}^{N} |slope(h_{f_{q_l}})| + \lambda_3 \sum_{l=1}^{N} SD(h_{a_l}[t+1] - h_{a_l}[t]) \tag{6}$$

where $\lambda_1$, $\lambda_2$, and $\lambda_3$ are tuning parameters that controls the trade-off between reconstruction error and the quality of disentanglement between aging and fluctuations. The goal of loss function is to minimize the reconstruction loss while ensuring the flatness of the fluctuation terms and smoothness of the aging term.

The output of PV-stGNN-PLR can be used to derive both estimated degradation pattern ($DP_{est}$) and global *PLR* defined in the Eq 7. $h_a$ directly provides the $DP_{est}$ for all PV systems in the PV fleet. It can also be used to derive the global *PLR* estimation for every system from its corresponding aging term. Instead of fitting a line to all elements in $h_{a_i}$ for a particular system and estimate *PLR* value from slope and intercept, global *PLR* is derived by calculating the percent of performance loss or gain between any element $v$ in $h_{a_i}$ and its corresponding one year apart element $v'$. Global *PLR* can be derived through:

$$PLR_{global} = \frac{1}{m} \sum_{i=1}^{m} \frac{v' - v}{v} \times 100\% \tag{7}$$

where $m$ is the total number of one year apart pairs in $h_{a_i}$.

## 2.2 Baseline methods

Traditional *PLR* estimation methods leverage both input timeseries and site-specific / satellite weather data. Meteorology data is used to construct power predictive models to predict the performance of PV systems at a given exposure environment which is then compared to the measured values to find performance loss. We compare the performance of PV-stGNN-PLR with the following trend analysis-based *PLR* estimation methods for degradation pattern extraction:

(1) <u>*6K*</u> [11]: The 6K model uses Plane of Array (POA) irradiance and module temperature, but models them as a fraction of standard irradiance and difference from standard temperature.

Additionally, this model requires a nameplate power input and will always predict the supposed nameplate power at Standard Test Conditions (STC).

(2) *PVUSA* [12]: PVUSA is a well-known physics-based *PLR* estimation model. The assumption of the model is that the current of a solar panel is a function of the irradiance and the voltage is a function of the irradiance and the module temperature, which is predicted by the ambient temperature and the wind speed.

(3) *XbX* [13]: The XbX model is a data-driven, multiple regression predictive model. The flexibility of this model enables non-linear, change point *PLR*. The X in the name refers to a given time step over which the power prediction model is built; a model built on a day of data would be Day-by-Day (DbD), while in Month-by-Month (MbM) modeling, data would be subset by months. The time step is chosen based on the condition of the data being modeled, and what modeling will be performed on the overall dataset.

(4) *XbX + UTC* [14]: The XbX + UTC is based on the XbX model by introducing a universal temperature correction (UTC) to produce a single temperature coefficient that can be used to convert to the desired representative temperature value.

(5) *QP-Aging-Detect* [15]: The QP-Aging-Detect is a mathematical model using Quadratic Programming (QP) to profile the long-term degradation in timeseries by decomposing them into the aging and fluctuation terms. For fair comparison, we remove the monotonic (strictly increasing or decreasing) constraint imposed on aging component. If we do not remove the monotonic constraint, QP-Aging-Detect will fail to extract the non-monotonic degradation patterns in the Case 2 data.

## 2.3 Datasets

We verified our model on three simulated PV datasets with "ground truth", consisting of 10-year power output timeseries from 100 inverters in 5 PV sites from Colorado, USA. Simulated PV Power output timeseries were derived from TMY Cyclic Weather using NSRDB Physical Solar Model v3 [24] and Sandia Array Performance Model from pvlib [25]. Each site contains spatially correlated inverters such that inverters in close proximity experience similar degradation severity. All the data was simulated in five clusters of twenty inverters, each with a ±4% noise in geo-spatial position and with a ±.2% noise in simulated power for each cluster. We also implemented our model on one real-world PV dataset without "ground truth", consisting of 5-year power output timeseries from 45 inverters from Florida, USA.

The three simulated datasets generated were controlled with configurable degradation patterns. The first dataset follows a linear degradation pattern, the second dataset follows a linear degradation pattern with a breakpoint (a change in the degradation rate from positive to negative slope at a certain point) after the first two years, and the third dataset follows a non-linear degradation pattern. The degradation patterns have been scaled such that for each degradation pattern, the nominal drop in performance over the length of timeseries is equivalent to a linear degradation rate with the supplied *PLR* value. For example, a system with a rated *PLR* of 2% per year would have a 80% performance compared to their nominal value for each degradation pattern.

## 2.4 Error measurements

We evaluated the *PLR* estimation errors using Euclidean Distance (ED) and Mean Absolute Percent Error (MAPE).

**Mean absolute percent error.** MAPE is defined as follows:

$$MAPE = \frac{1}{N \times T} \sum_{L=1}^{N} \sum_{J=1}^{T} \frac{|DP_{est_{L_J}} - DP_{real_{L_J}}|}{|DP_{real_{L_J}}|} \times 100\% \qquad (8)$$

where $DP_{est_{L_J}}$ is the $J - th$ coefficient of $DP_{est}$ for node $L$.

Smaller values of ED and MAPE indicate better *PLR* estimations, as they represent a smaller estimation error between the $DP_{est}$ and $DP_{real}$.

**Euclidean distance.** Reporting a single *PLR* for the entire system does not capture the system's performance dynamics over time. Euclidean distance measures spatial separation between RDP (real degradation patterns) and EDP (estimated degradation patterns) between two vectors, in this case the degradation patterns, which allows us to judge the similarity of the vectors, not just difference between each pairwise value. A novel way to quantify how well a model can estimate *PLR* was developed using Euclidean Distance. It consists of two steps: (1) Rescaling: Given a real degradation pattern ($DP_{real}$) and an estimated degradation pattern ($DP_{est}$), we rescale every data point in $DP_{real}$ and $DP_{est}$ by dividing them by their respective first values, and (2) Error Calculation: We calculate ED between scaled $DP_{real}$ and $DP_{est}$.

## 3 Results

Four experiments were conducted to verify the performance of PV-stGNN-PLR compared with state-of-the-art baselines in terms of accuracy and case analysis of *PLR* trends. The first three experiments were conducted using simulated PV data while the last experiment was conducted using real-world PV data. The simulated PV data represented three distinct aging trends, including two non-linear trends, designated by different "Cases". Case 1 data has a linear degradation pattern, Case 2 data exhibits a upward trend for two years then has a linear degradation for eight years, Case 3 data has a non-linear degradation trend. After benchmarking model performance on simulated data, PV-stGNN-PLR was applied to a fleet of 45 PV systems located throughout Florida.

### 3.1 Evaluation of *PLR* degradation estimation errors

To evaluate the *PLR* degradation pattern estimation accuracy, we compared MAPE and ED between the $DP_{real}$ and $DP_{est}$. Table 1 presents *PLR* degradation pattern estimation errors of PV-stGNN-PLR and baselines over three datasets, each exhibiting a different type of degradation pattern: linear, linear with breakpoint, and non-linear.

**Table 1. *PLR* degradation pattern estimation error comparisons (simulated data).** Best performance is highlighted in bold font (PV-stGNN-*PLR*-NF: PV-stGNN-*PLR* without flatness regularization).

| Models | Datasets | | | | | |
|---|---|---|---|---|---|---|
| | Case 1: Linear | | Case 2: Linear w. Breakpoint | | Case 3: Non-linear | |
| | MAPE (%) | Euclidean Distance | MAPE (%) | Euclidean Distance | MAPE (%) | Euclidean Distance |
| 6K | 11.97 | 0.4100 | 11.11 | 0.4235 | 12.14 | 0.4102 |
| PVUSA | 4.66 | 0.1666 | 4.59 | 0.1780 | 4.71 | 0.1661 |
| XbX | 4.21 | 0.1467 | 4.20 | 0.1609 | 4.18 | 0.1440 |
| XbX+UTC | 1.27 | 0.0515 | 1.22 | 0.0454 | 1.21 | 0.0443 |
| QP-Aging-Detect | 2.38 | 0.0712 | 1.18 | 0.0432 | 2.57 | 0.0730 |
| PV-stGNN-PLR | **0.40** | **0.0150** | **0.96** | **0.0362** | **1.06** | **0.0381** |
| PV-stGNN-PLR-NF | 2.84 | 0.0939 | 1.46 | 0.0523 | 2.33 | 0.0761 |

We observe that the PV-stGNN-PLR model achieves the best overall *PLR* degradation pattern estimation measured by MAPE and ED. Compared to the best performed baseline XbX +UTC, PV-stGNN-PLR achieves a reduction of 33.9% and 35.1% on average in MAPE and ED. Even in the more challenging degradation pattern cases 2 and 3, PV-stGNN-PLR continues to outperform the other baselines significantly.

### 3.2 Analysis of PV-stGNN-PLR performance

We conducted case studies to validate the quality of disentanglement between aging and fluctuation terms by PV-stGNN-PLR. We also provided detailed visualization of estimated degradation patterns derived from PV-stGNN-PLR and baselines compared to real degradation patterns.

Fig 2, illustrates extracted aging and fluctuation terms of a PV system under linear with breakpoint and non-linear degradation patterns. From this figure, we observe that PV-stGNN-PLR achieves a clear disentanglement between aging and fluctuation terms. The aging term successfully captures the initial upward trend in the Case 2 degradation pattern. Fluctuation term captures annual seasonality and noises from the input timeseries.

In addition, we compared $DP_{est}$ extracted by PV-stGNN-PLR and baselines with the ground truth $DP_{real}$. Fig 3, shows PV-stGNN-PLR can best recover real degradation patterns from input timeseries compared to baselines. We can see $DP_{est}$ extracted by PV-stGNN-PLR is the closest to $DP_{real}$ followed by XbX+UTC, which is consistent with results shown in Table 1.

Fig 4, shows an example of a degradation pattern extracted by PV-stGNN-PLR for a PV inverter with a ten-years-long power output timeseries.

### 3.3 Evaluation of regularization terms on model performance

To verify how regularization imposed on the learning objective improves degradation pattern estimation, we conducted an ablation analysis to study the impacts of flatness regularization across all three degradation patterns. After removing flatness regularization from our learning objective, we found that PV-stGNN-PLR with flatness regularization reduces MAPE and ED by 58.10% and 54.92% on average compared to PV-stGNN-PLR without flatness regularization. This observation is consistent with its counterpart in Fig 3.

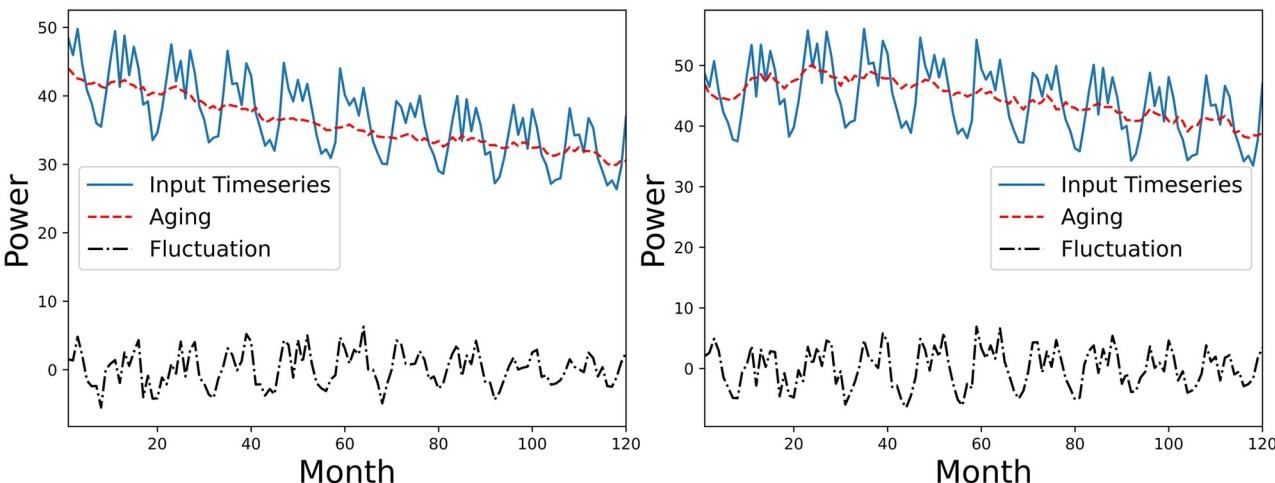

**Fig 2. Example of extracted aging (in red) and fluctuation terms (in black) by PV-stGNN-PLR for a PV system (simulated data).** Left Figure: Case 3 Simulated Degradation, non-linear degradation. Right Figure: Case 2 Simulated Degradation, linear with breakpoint degradation.

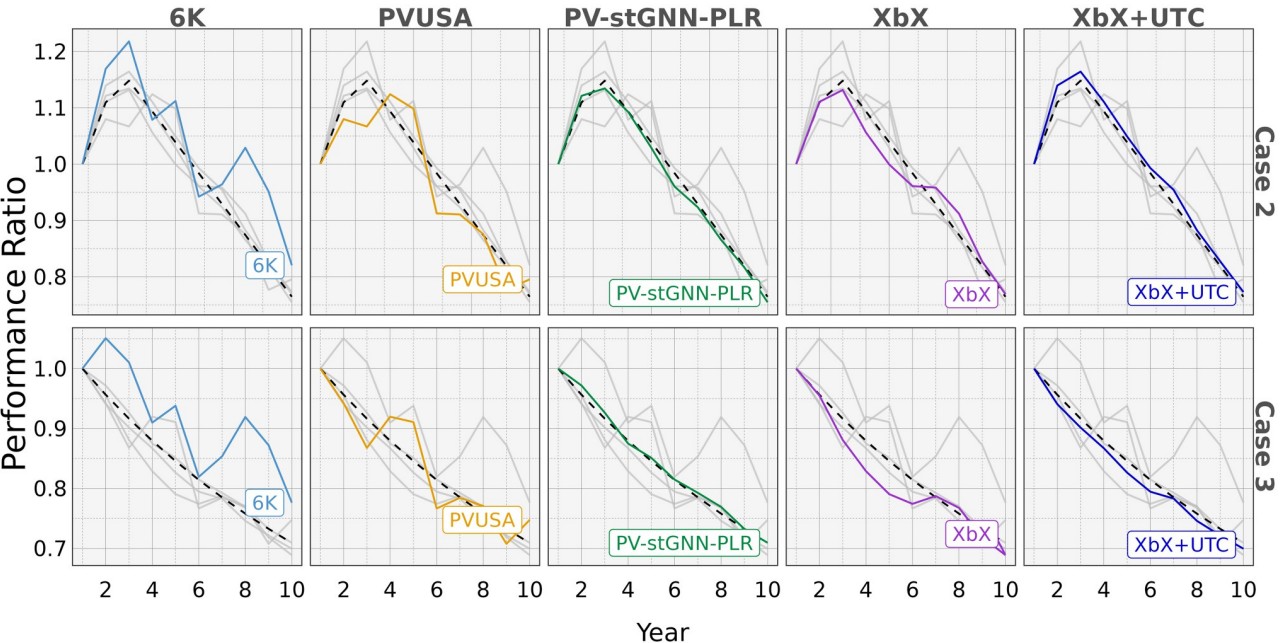

**Fig 3. Comparison of a real degradation pattern with estimated degradation pattern extracted by PV-stGNN-PLR and baselines (simulated dataset) for Case 2 and Case 3 degradation patterns.** Each model is highlighted over the other models that have been grayed out for better comparison.

## 3.4 Evaluation of PV-stGNN-PLR on real-world PV datasets

We have applied PV-stGNN-PLR to real-world PV datasets consisting of 5-years power timeseries from 45 PV systems. As shown in Fig 5, PV-stGNN-PLR decomposes PV timeseries from two systems into their corresponding aging and fluctuation terms. PV-stGNN-PLR achieves a clear disentanglement between aging and fluctuation terms in both cases. The fluctuation term is non-stationary, moving on average consistently close to zero, which captures

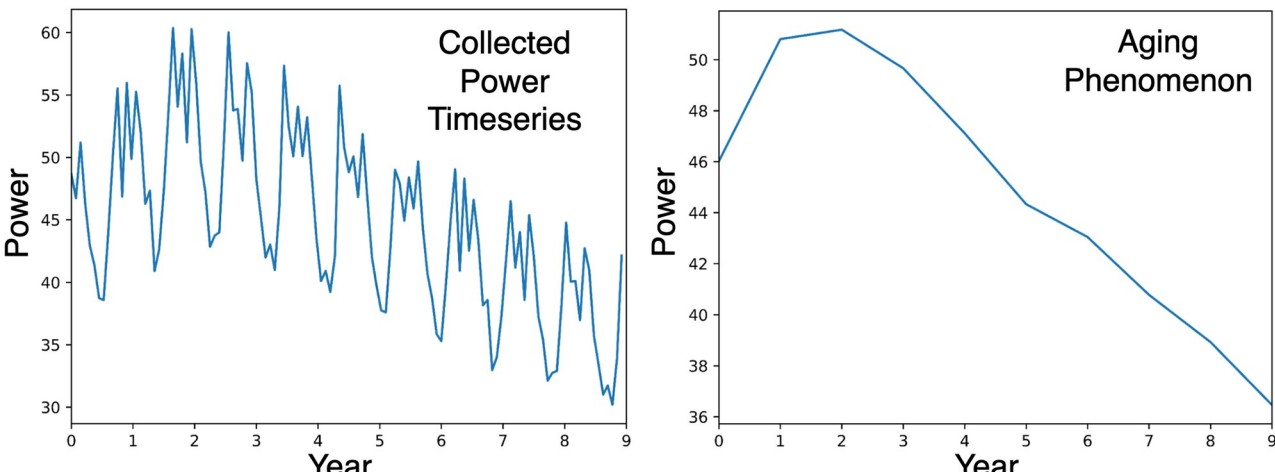

**Fig 4. Example of a PV system with the non-monotonic degradation pattern (simulated data).**

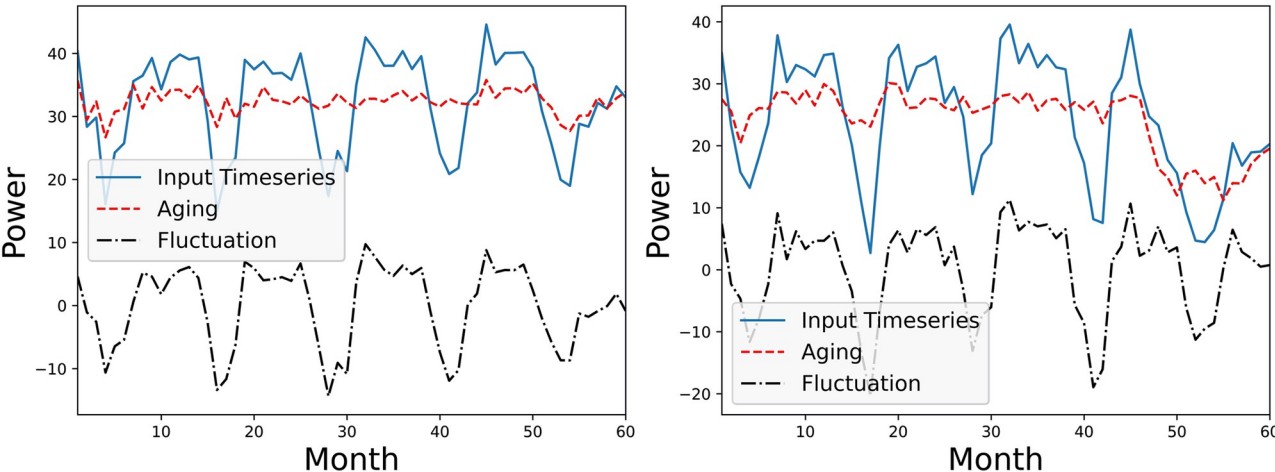

**Fig 5. Example of extracted aging (in red) and fluctuation terms (in black) by PV-stGNN-PLR for two PV systems (real-world data).**

the seasonalities and noises within the timeseries. Aging terms reflect the estimated degradation pattern. We observe that the aging term successfully detects the performance drift on the right part of Fig 5. Since it is impossible to obtain the actual degradation patterns and *PLR* for real-world PV systems, we cannot conduct degradation pattern estimation error analysis on them.

## 4 Discussion

The findings of our study highlight the potential of PV-stGNN-PLR to accurately estimate *PLR* degradation patterns in large fleets of PV systems. The evaluation metrics, MAPE (Mean Absolute Percentage Error) and ED (Euclidean Distance), are used to compare the performance of PV-stGNN-PLR with baselines. Our results in Table 1 demonstrate that the design of our graph autoencoder and the learning objective in PV-stGNN-PLR effectively separate the aging and fluctuation components in the PV dataset to derive accurate degradation pattern estimation. Specifically, PV-stGNN-PLR successfully captures the linear "up" trend in the first two years followed by a subsequent linear "down" trend until the end of the dataset's time span in Case 2 degradation pattern (Fig 4). This indicates that the PV-stGNN-PLR models accurately capture the aging and degradation patterns exhibited by PV systems over time, providing valuable insights into the performance dynamics of the systems.

To further validate the quality of disentanglement between aging and fluctuation terms by PV-stGNN-PLR, case study in Fig 2 provides an example of the extracted aging and fluctuation terms for a PV system exhibiting linear with a break-point and non-linear degradation patterns. This figure demonstrates the clear disentanglement achieved by PV-stGNN-PLR, where the aging term captures the initial upward trend in the degradation patterns, while the fluctuation term captures the seasonal variations and noise in the input timeseries. This confirms the effectiveness of the graph autoencoder design and learning objective of PV-stGNN-PLR in separating aging and fluctuation in PV datasets for complex degradation patterns.

Moreover, the extracted degradation patterns ($DP_{est}$) by PV-stGNN-PLR and baselines are compared with the ground truth real degradation patterns ($DP_{real}$). Fig 3 shows that PV-stGNN-PLR provides the closest estimation to the $DP_{real}$, followed by XbX+UTC, consistent with the results shown in Table 1. Additionally, Fig 2 presents two examples of the degradation

pattern extracted by PV-stGNN-PLR for a PV inverter over a ten-year period. It demonstrates PV-stGNN-PLR's ability to capture both types of non-linearity in degradation pattern extraction.

Application on 45 real world systems PV-stGNN-PLR ability to capture both long term trend and instantaneous performance drift. This module could not only be used to estimate long term system performance but also to inform instantaneous maintenance or replacement of damaged assets. The extracted aging trends can also be used to make accurate financial models on when modules should reach their end of life.

Overall, our results highlight the accuracy, disentanglement capability, and effectiveness of PV-stGNN-PLR in estimating *PLR* degradation patterns for PV systems in a large PV fleet. Accurate estimations of performance loss as a timeseries object in lieu of a linear approximations enable precise determinations of production capacity and system health. PV-stGNN-PLR has the potential to become a valuable tool for understanding and predicting the performance dynamics of PV systems, which can aid in decision-making for PV manufacturers and enhance the financial feasibility of PV energy solutions.

## 5 Conclusion

We have proposed PV-stGNN-PLR, a novel neural network based timeseries decomposition method that adopts paralleled graph autoencoder modules to decompose input PV power timeseries into aging (performance loss) and fluctuation terms for long-term trend (*PLR*) analysis. Each module exploits spatial coherence from neighboring PV systems and temporal correlations within-series to perform aging and fluctuation extractions for PV fleet-level analysis. The loss function of PV-stGNN-PLR ensures the clear disentanglement between aging and fluctuation through smoothness (imposed on aging) and flatness (imposed on fluctuation) regularizations. These operations are conducted without the need for input site-specific meteorology data, allowing for the determination of *PLR* without curating additional data flows.

Our experimental study has verified that PV-stGNN-PLR outperforms existing state-of-the-art *PLR* estimation methods over PV datasets with different degradation patterns: linear, linear with breakpoint, and non-linear. PV-stGNN-PLR achieves a reduction of 33.9% and 35.1% on average in Mean Absolute Percent Error and Euclidean Distance given the best baseline XbX+UTC. In particular, it outperforms the other baselines significantly in more challenging signal decomposition problems; linear with breakpoint and non-linear degradation patterns. PV-stGNN-PLR represents a state of the art signal decomposition method that can accurately decompose trends with incredibly low signal to noise ratios, out performing physics-based and state of the art methods. The accuracy of PV-stGNN-PLR allows for the precise calculation of system degradation which better informs the LCOE of PV technologies.

## Acknowledgments

This work made use of the High Performance Computing Resource in the Core Facility for Advanced Research Computing at Case Western Reserve University.

## Author Contributions

**Conceptualization:** Yangxin Fan, Raymond Wieser, Yinghui Wu, Laura S. Bruckman, Roger H. French.

**Data curation:** Yangxin Fan, Raymond Wieser.

**Formal analysis:** Yangxin Fan.

**Funding acquisition:** Yinghui Wu, Laura S. Bruckman, Roger H. French.

**Investigation:** Yangxin Fan.

**Methodology:** Yangxin Fan, Raymond Wieser, Yinghui Wu.

**Project administration:** Xuanji Yu, Yinghui Wu, Laura S. Bruckman, Roger H. French.

**Resources:** Yangxin Fan, Xuanji Yu, Yinghui Wu, Laura S. Bruckman, Roger H. French.

**Software:** Yangxin Fan.

**Supervision:** Xuanji Yu, Yinghui Wu, Laura S. Bruckman, Roger H. French.

**Validation:** Yangxin Fan.

**Visualization:** Yangxin Fan, Raymond Wieser.

**Writing – original draft:** Yangxin Fan.

**Writing – review & editing:** Yangxin Fan, Raymond Wieser, Yinghui Wu, Laura S. Bruckman, Roger H. French.

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
