## [Decision Letter · Decision Letter 0]

16 Oct 2023

PONE-D-23-29182Using spatio-temporal graph neural network to estimate fleet-level photovoltaic performance loss ratePLOS ONE

Dear Dr. Wieser,

Thank you for submitting your manuscript to PLOS ONE. After careful consideration, we feel that it has merit but does not fully meet PLOS ONE’s publication criteria as it currently stands. Therefore, we invite you to submit a revised version of the manuscript that addresses the points raised during the review process. In particular, one reviewer provides a number of significant suggestions, including the utilization of specific methodologies to handle the seasonality of the data. It is thus recommended that you take into account these recommendations when revising the manuscript.

We look forward to receiving your revised manuscript.

Kind regards,

Emanuele Crisostomi, PhD

Academic Editor

PLOS ONE

Journal Requirements:

4. Thank you for stating the following in the Acknowledgments Section of your manuscript: "This material is based upon work supported by the U.S. Department of Energy’s Office of Energy

Efficiency and Renewable Energy (EERE) under Solar Energy Technologies Office (SETO) Agreement

Number DE-EE0009353. The views expressed herein do not necessarily represent the views of the U.S.

Department of Energy or the United States Government. This work made use of the High Performance

Computing Resource in the Core Facility for Advanced Research Computing at Case Western Reserve

University."

Please remove any funding-related text from the manuscript and let us know how you would like to update your Funding Statement. Currently, your Funding Statement reads as follows: "Y.F, R.W, X.Y, Y.W, L.S.B,F.H.F

This material is based upon work supported by the U.S. Department of Energy’s Office of Energy Efficiency and Renewable Energy (EERE) under Solar Energy Technologies Office (SETO) Agreement Number DE-EE0009353.  

he funders had no role in study design, data collection and analysis, decision to publish, or preparation of the manuscript."

6. We note that Figure 1 in your submission contain [map/satellite] images which may be copyrighted. All PLOS content is published under the Creative Commons Attribution License (CC BY 4.0), which means that the manuscript, images, and Supporting Information files will be freely available online, and any third party is permitted to access, download, copy, distribute, and use these materials in any way, even commercially, with proper attribution. For these reasons, we cannot publish previously copyrighted maps or satellite images created using proprietary data, such as Google software (Google Maps, Street View, and Earth). For more information, see our copyright guidelines: http://journals.plos.org/plosone/s/licenses-and-copyright.

Reviewers' comments:

Reviewer's Responses to Questions

**Comments to the Author**

1. Is the manuscript technically sound, and do the data support the conclusions?

Reviewer #1: Yes

Reviewer #2: Yes

2. Has the statistical analysis been performed appropriately and rigorously? 

Reviewer #1: Yes

Reviewer #2: Yes

3. Have the authors made all data underlying the findings in their manuscript fully available?

Reviewer #1: Yes

Reviewer #2: Yes

4. Is the manuscript presented in an intelligible fashion and written in standard English?

Reviewer #1: Yes

Reviewer #2: Yes

5. Review Comments to the Author

Reviewer #1: The manuscript entitled, “Using spatio-temporal graph neural network to estimate fleet-level photovoltaic performance loss rate” displays Spatio-Temporal Graph Neural Network for fleet-level PLR estimation (PV-stGNN-PLR) which decomposes the power time series data into aging and fluctuation components, utilizing the aging component to determine PLR. PV-stGNN-PLR exploits spatial and temporal coherence to derive PLR estimation for all systems in a fleet and imposes flatness and smoothness regularization in loss function to ensure the successful disentanglement between aging and fluctuation. There exist a few tribulations stated below that need to be corrected.

1. The authors have mentioned PLR in the ‘Introduction’ section of the manuscript which is supported by reference [5] in the manuscript. The author must express the formula for PLR in the present case as expressed by Deceglie et al. https://doi.org/10.1002/solr.202300196

2. What is POA in 6K model?

3. The authors have used several acronyms in the manuscript which need to be defined in the manuscript.

4. MAPE(%) is found least for PV-stGNN-PLR in all the three discussed cases in the manuscript. However, this loss function of PV-stGNN-PLR consists of a global reconstruction error with additional constraints applied individually to the fluctuation and aging terms. Is this is the reason for achieving low MAPE(%). Please justify!

5. Since your data contains seasonality, did you try Holt-winters exponential smoothing? It will be great to handle the time series data containing a seasonality.

6. Can you please explain why did you choose Euclidean distance and MAPE to evaluate PLR estimation errors? Did you consider any other Time Series Error metrics?

7. The alignment of the text in the manuscript is not justified.

8. Define ‘mqlj’ in the equation (3).

9. There are several grammatical errors in the manuscript.

10. References are randomly collected without any standard format style such as APA, Chicago. MLA, etc.

Reviewer #2: I really appreciate the research carried out in the proposed article and I wish to see it published. The paper is very well written and presented. I would be glad to see your future publications in this field.

6. PLOS authors have the option to publish the peer review history of their article (what does this mean?). If published, this will include your full peer review and any attached files.

Reviewer #1: **Yes: **Shalini Tripathi

Reviewer #2: No

---

## [Author Response · Author response to Decision Letter 0]

20 Nov 2023

Reviewer 1:

The authors have mentioned PLR in the ‘Introduction’ section of the manuscript which is supported by reference [5] in the manuscript. The author must express the formula for PLR in the present case as expressed by Deceglie et al.

The authors have reviewed the manuscript suggested by Reviewer #1 and feel like the equation provided in this manuscript (Eq 7) is equivalent to the one expressed by Deceglie et al.

What is POA in 6K model?

We have updated our manuscript to define the Plane of Array (POA) Irradiance in the body of the text. 

The authors have used several acronyms in the manuscript which need to be defined in the manuscript.

The authors have read through the manuscript again and added any undefined acronyms

MAPE(%) is found least for PV-stGNN-PLR in all the three discussed cases in the manuscript. However, this loss function of PV-stGNN-PLR consists of a global reconstruction error with additional constraints applied individually to the fluctuation and aging terms. Is this is the reason for achieving low MAPE(%). Please justify!

Yes, our carefully designed loss function ensures that PV-stGNN-PLR can accurately extract the complex (non-linear and/or non-monotonic) performance loss patterns of PV systems from aging terms. The fluctuation terms extract multi-level seasonal patterns for each PV system constrained by the topological information captured by the spatiotemporal graph and global reconstruction error guarantees that the extracted aging terms match with corresponding PV systems.

Since your data contains seasonality, did you try Holt-winters exponential smoothing? It will be great to handle the time series data containing a seasonality.

No, holt-winters exponential smoothing can only be used to extract linear, linear with damping, and exponential trend [1]. Holt-Winters is another commonly applied model for PLR estimation, but was not implemented in the scope of this study. It fails to capture the more complex non-linear and/or non-monotonic performance loss patterns in PV systems.However, our design PV-stGNN-PLR, an unsupervised machine learning method, can learn the complex non-linear PLR of PV systems.

[1]: Kalekar, Prajakta S. "Time series forecasting using holt-winters exponential smoothing." Kanwal Rekhi school of information Technology 4329008.13 (2004): 1-13.

Can you please explain why did you choose Euclidean distance and MAPE to evaluate PLR estimation errors? Did you consider any other Time Series Error metrics?

Yes, other metrics MAE, MSE, and RMSE are also found in literature to report PLR estimation errors. However, these values are in same units as the original data, which makes them harder to compare the accuracy across data with different scales. MAPE, on the other hand, express the error as a percentage of actual values making it easier to compare prediction errors across PV systems. Hence, if MAPE is 1.5%, this means the average prediction error is 1.5% of actual values. Euclidean distance measures spatial separation between RDP (real degradation patterns) and EDP (estimated degradation patterns) which are both sequences and can be naturally considered as two data points in a high-dimension space. We have updated the methodology with some clarifications of Euclidean Distance and its importance in our model.

The alignment of the text in the manuscript is not justified.

The authors have used the supplied manuscript template provided by PLOSone. We have not edited the document setup parameters. 

Define ‘mqlj’ in the equation (3).

 The authors have clarified the element Mqlj in the body of the manuscript.

There are several grammatical errors in the manuscript.

The authors have read through the manuscript and checked for grammatical errors, correcting them as fit. 

References are randomly collected without any standard format style such as APA, Chicago. MLA, etc.

The authors have followed the instructions on PLOSone guidelines on the formatting of the references. They are formatted correctly as per the instructions on PLOSone.

Reviewer #2

I really appreciate the research carried out in the proposed article and I wish to see it published. The paper is very well written and presented. I would be glad to see your future publications in this field.

 The authors appreciate the work reviewers due to ensure a quality submission.

---

## [Decision Letter · Decision Letter 1]

4 Jan 2024

Using spatio-temporal graph neural networks to estimate fleet-wide photovoltaic performance degradation patterns

PONE-D-23-29182R1

Dear Dr. Wieser,

We’re pleased to inform you that your manuscript has been judged scientifically suitable for publication and will be formally accepted for publication once it meets all outstanding technical requirements.

Kind regards,

Emanuele Crisostomi, PhD

Academic Editor

PLOS ONE

Additional Editor Comments (optional):

Reviewers' comments:

Reviewer's Responses to Questions

**Comments to the Author**

1. If the authors have adequately addressed your comments raised in a previous round of review and you feel that this manuscript is now acceptable for publication, you may indicate that here to bypass the “Comments to the Author” section, enter your conflict of interest statement in the “Confidential to Editor” section, and submit your "Accept" recommendation.

Reviewer #3: All comments have been addressed

Reviewer #4: All comments have been addressed

2. Is the manuscript technically sound, and do the data support the conclusions?

Reviewer #3: (No Response)

Reviewer #4: Yes

3. Has the statistical analysis been performed appropriately and rigorously? 

Reviewer #3: (No Response)

Reviewer #4: Yes

4. Have the authors made all data underlying the findings in their manuscript fully available?

Reviewer #3: (No Response)

Reviewer #4: Yes

5. Is the manuscript presented in an intelligible fashion and written in standard English?

Reviewer #3: (No Response)

Reviewer #4: Yes

6. Review Comments to the Author

Reviewer #3: (No Response)

Reviewer #4: The authors have adequately addressed comments of reviewers raised in a previous round of review. I have no issues to raise in this version of the manuscript. The paper is well-written and adequately describe the methods and results. The proposed algorithm can be applied in a wide range of tasks where separating aging and fluctuation is required.

7. PLOS authors have the option to publish the peer review history of their article (what does this mean?). If published, this will include your full peer review and any attached files.

Reviewer #3: **Yes: **Dong Chen

Reviewer #4: No
